

# *Dugong dugon* feeding in tropical Australian seagrass meadows: implications for conservation planning

Samantha J. Tol[1,2], Rob G. Coles[1] and Bradley C. Congdon[2]

[1] Centre for Tropical Water and Aquatic Ecosystem Research, James Cook University, Cairns, Queensland, Australia
[2] College of Marine and Environmental Sciences, James Cook University, Cairns, Queensland, Australia

## ABSTRACT

Dugongs (*Dugong dugon*) are listed as vulnerable to extinction due to rapid population reductions caused in part by loss of seagrass feeding meadows. Understanding dugong feeding behaviour in tropical Australia, where the majority of dugongs live, will assist conservation strategies. We examined whether feeding patterns in intertidal seagrass meadows in tropical north-eastern Australia were related to seagrass biomass, species composition and/or nitrogen content. The total biomass of each seagrass species removed by feeding dugongs was measured and compared to its relative availability. Nitrogen concentrations were also determined for each seagrass species present at the sites. Dugongs consumed seagrass species in proportion to their availability, with biomass being the primary determining factor. Species composition and/or nitrogen content influenced consumption to a lesser degree. Conservation plans focused on protecting high biomass intertidal seagrass meadows are likely to be most effective at ensuring the survival of dugong in tropical north-eastern Australia.

## INTRODUCTION

Dugongs (*Dugong dugon*) are the last surviving species within the family Dugongidae (*Grech, Sheppard & Marsh*, *2011*; *Marsh & Lefevbre*, *1994*) and are listed as 'vulnerable to extinction' by the IUCN Red List of Threatened Species (*IUCN*, *2011*). Dugong diet consists predominantly of seagrass and they are restricted in range to tropical and subtropical locations of the Indo-west Pacific where shallow seagrass meadows are common (*Marsh et al.*, *1982*; *Marsh, O'shea & Reynolds III*, *2011*). Throughout most of this distribution they occur as small relatively isolated populations (*Allen, Marsh & Hodgson*, *2004*), with the only substantial populations being found in northern Australian waters (*Marsh, Grech & Hagihara*, *2012*; *Marsh & Lefevbre*, *1994*; *Marsh, O'shea & Reynolds III*, *2011*). Since the 1960s, dugong populations along the east coast of Queensland, Australia, have declined by 95 percent (*Marsh et al.*, *2005*; *Marsh & Lefevbre*, *1994*), leading to calls for the species status to be up listed to critically endangered (*Marsh et al.*, *2005*; *Marsh, O'shea & Reynolds III*, *2011*). In eastern Queensland much of the decline has occurred since 2005,

Corresponding author
Samantha J. Tol,
samantha.tol@jcu.edu.au

and has been attributed to substantial reductions in seagrass availability associated with recent extreme weather events (*McKenna et al.*, *2015*; *McKenzie, Collier & Waycott*, *2012*; *Sobtzick et al.*, *2012*).

In Australia, studies of dugong feeding in the subtropical, subtidal meadows of south-east Queensland suggest that dugong feeding preferences are based on the nutritional quality and digestibility of seagrass as a food source (*Marsh, O'shea & Reynolds III*, *2011*; *Preen*, *1995*; *Sheppard et al.*, *2010*). These studies observed preferences for *Halophila ovalis* and *Halodule uninervis* and concluded that these species are targeted because of their greater nitrogen content, combined with a general preference for low biomass strands, due to lower concentrations of fibre (*Lanyon*, *1991*; *Mellors, Waycott & Marsh*, *2005*; *Preen*, *1995*; *Sheppard, Lawler & Marsh*, *2007*). This is consistent with nitrogen being a major limiting nutrient for all Sirenians, including dugongs (*Lanyon*, *1991*). Research comparing nutrient content also found that intertidal seagrass plants have higher levels of starch and are more digestible than subtidal plants (*Sheppard et al.*, *2008*; *Sheppard, Lawler & Marsh*, *2007*). Combined, these studies suggest that intertidal seagrass meadows with species of high available nitrogen should be preferred feeding grounds (*Preen*, *1998*; *Sheppard et al.*, *2010*). In contrast, studies of dugong feeding in other tropical regions regularly observe feeding across all species present, with the exception of *Enhalus acoroides* (*Adulyanukosol & Poovachiranon*, *2006*; *André, Gyuris & Lawler*, *2005*; *Aragones*, *1994*; *De Iongh et al.*, *2007*). The differences in dugong feeding behaviour observed at the edge of their range in south-east Queensland may be due to seagrass biodiversity being lower, combined with dietary requirements for living in colder waters (*Preen*, *1992*; *Preen*, *1995*). Therefore, the applicability of findings from south-east Queensland to intertidal feeding areas in tropical Australia, or to other tropical feeding grounds is unclear.

Many factors may influence dugong feeding choice. Alterations in feeding preferences and behaviour could occur if the animal is under stress, such as from temperature at the edge of their distributional range, or hunting pressures (*Anderson*, *1994*; *Anderson*, *1998*; *Anderson & Birtles*, *1978*; *Brownell et al.*, *1981*; *Marsh, O'shea & Reynolds III*, *2011*; *Wirsing, Heithaus & Dill*, *2007*). Other important factors include the physical characteristics of the marine environment such as depth, sediment type, water temperature and water currents, or biological factors such as the seagrass species type, biomass above and below ground, digestibility, nutrient content and the age of the plants (*Aragones et al.*, *2006*; *Marsh, O'shea & Reynolds III*, *2011*; *Preen*, *1995*). Theoretically, dugong feeding behaviour is thought to follow an optimal foraging strategy, in which feeding site selection is based on maximum energy gained with minimal energy expended (*Aragones et al.*, *2006*; *Preen*, *1995*; *Sheppard, Lawler & Marsh*, *2007*). Adding to this complexity, is that feeding location and the timing of feeding may also be influenced by external factors such as the possibility of stranding, presence of predators, human disturbance, seasonal changes or simply familiarity with the area and its seagrass meadow history (*Marsh, O'shea & Reynolds III*, *2011*). Understanding the reason dugongs choose to feed in some locations and not others, and/or the importance of specific seagrass species or meadows, is important for developing appropriately targeted conservation of high quality dugong feeding habitats.

**Table 1  Description of seagrass meadow study sites identifying sediment type, species present and average percent cover (m²) and biomass (g/DW m²); percent cover and biomass was measured over an area ranging from 1,000–1,500 m² over a 4-month period.** Data collected during March to July 2012 of six intertidal seagrass meadows along the Great Barrier Reef in the north-east Queensland, Australia.

| Seagrass meadow | Sediment | Species present | Average species % Cover (m²) | SE | Average biomass (g/DW m²) | SE |
|---|---|---|---|---|---|---|
| Cooya Beach | Mud and sand | *Enhalus acoroides* | NR | | NR | |
| | | *Halodule uninervis* | NR | | NR | |
| | | *Halophila ovalis* | NR | | NR | |
| | | *Zostera muelleri* | NR | | NR | |
| Yule Point | Sand | *Halodule uninervis* | NR | | NR | |
| | | *Halophila ovalis* | NR | | NR | |
| Double Island, Reef Flat | Coarse sand, coral rubble and mud | *Cymodocea spp.* | 4.419 | ±1.327 | 0.239 | ±0.051 |
| | | *Halodule uninervis* | 73.566 | ±2.709 | 0.586 | ±0.043 |
| | | *Halophila ovalis* | 20.559 | ±2.558 | 0.124 | ±0.016 |
| | | *Syringodium isoetifolium* | 0.118 | ±0.060 | 0.060 | ±0.022 |
| | | *Thalassia hemprichii* | 0.831 | ±0.298 | 0.179 | ±0.037 |
| Cape Pallarenda | Sand and mud | *Halodule uninervis* | 38.038 | ±2.144 | 1.298 | ±0.088 |
| | | *Halophila ovalis* | 61.962 | ±2.144 | 0.636 | ±0.018 |
| Cockle Bay, Magnetic Island | Sand, coral rubble and mud | *Cymodocea spp.* | 2.733 | ±0.456 | 0.229 | ±0.033 |
| | | *Halodule uninervis* | 1.698 | ±0.418 | 0.363 | ±0.087 |
| | | *Halophila ovalis* | 95.276 | ±0.570 | 0.797 | ±0.014 |
| | | *Thalassia hemprichii* | 0.293 | ±0.126 | 0.318 | ±0.111 |
| Bowling Green Bay | Mud and sand | *Halodule uninervis* | NR | | NR | |
| | | *Halophila ovalis* | NR | | NR | |
| | | *Zostera muelleri* | NR | | NR | |

**Notes.**

NR, Not recorded.

In the present study, we examine a sub-set of potential factors influencing the selection of seagrass species by dugongs feeding in intertidal seagrass meadows in tropical north Queensland, so as to improve our understanding of the relationship between dugong and their seagrass food. These factors include seagrass species presence, nitrogen content and relative biomass.

## MATERIALS & METHODS

### Study site

Six mixed species coastal intertidal seagrass meadows in the Great Barrier Reef (GBR), north Queensland, were selected based on the number of seagrass species present, a lack of direct human disturbance, and that dugong feeding has been historically recorded at these sites (*Davies & Rasheed*, *2016*; *McKenzie, Collier & Waycott*, *2014*) (Table 1). Three meadows were located near Cairns and three in the Townsville region (Fig. 1). The Cairns sites included a reef platform at Double Island located two kilometres offshore from

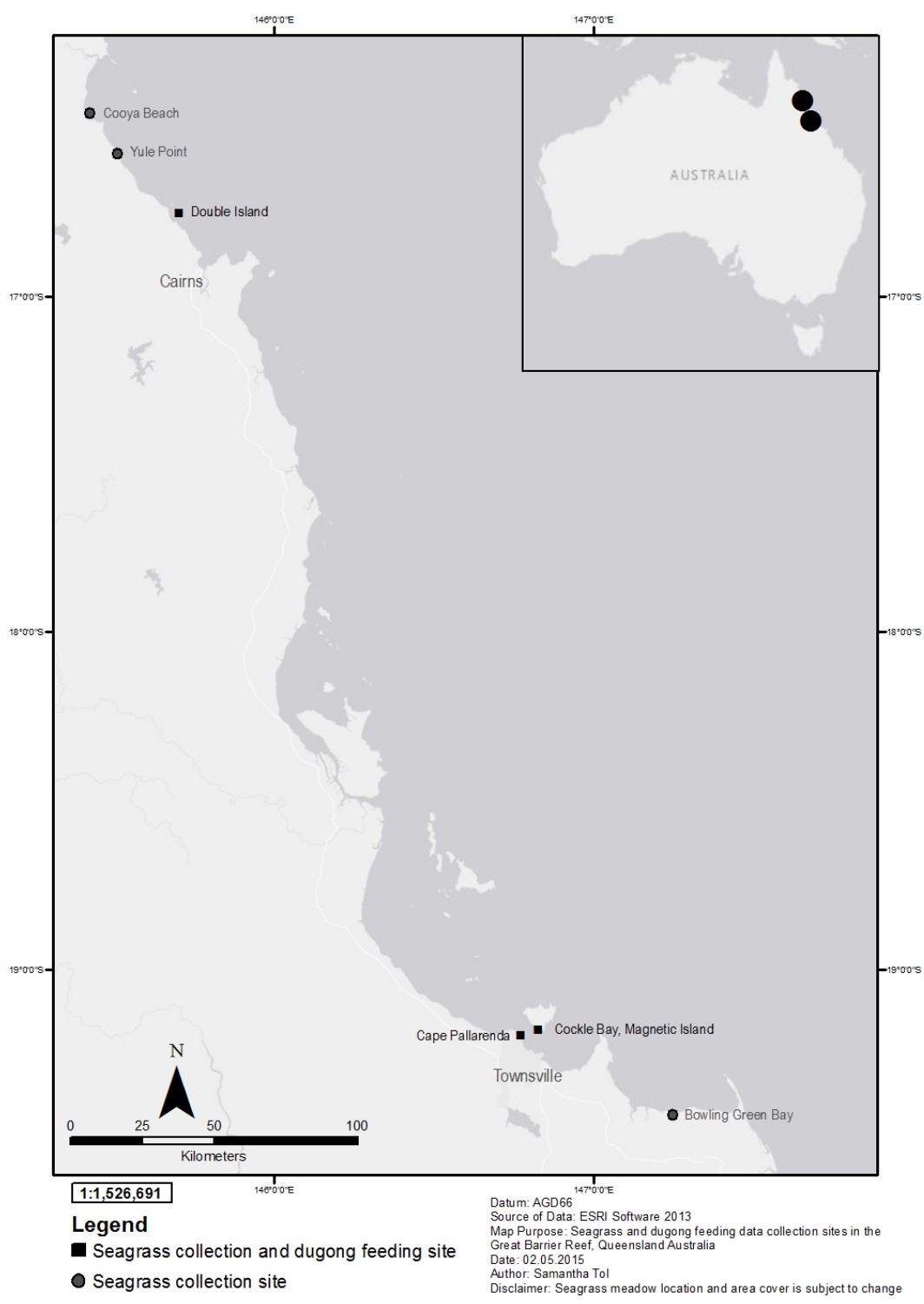

**Figure 1** **Seagrass collection and dugong feeding observation sites in the Great Barrier Reef, north-east Queensland Australia; locator map of Australia with Townsville and Cairns highlighted.** Source: ESRI software.

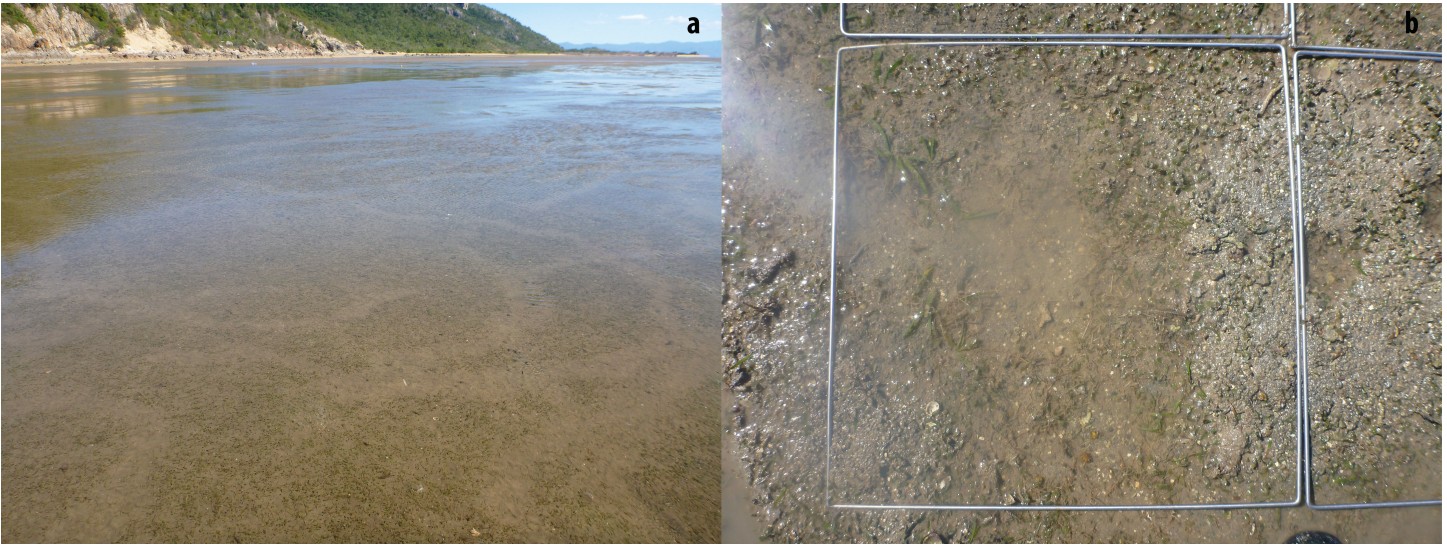

**Figure 2** (A) Dugong feeding trails in a mixed species intertidal seagrass meadow at low tide taken at Cape Pallarenda, Australia on the 02/07/2012; and (B) a dugong feeding trail through *Cymodocea rotundata* and *Halophila ovalis* seagrass at Cockle Bay, Magnetic Island Australia taken on the 29/07/2012.

the Cairns coastal beaches, Yule Point beach and Cooya beach north of Cairns. The Townsville sites were at Cockle Bay on Magnetic Island located seven kilometres off the coast of Townsville, Shelley Beach at Cape Pallarenda located north-west of the Townsville harbour, and Bowling Green Bay, an intertidal flat south-east of Townsville. Data on seagrass characteristics were collected from all six meadows and dugong feeding data were collected from three meadows; Cockle Bay, Cape Pallarenda and Double Island. The meadows studied, and meadows in the vicinity of the study sites, had consistent seagrass cover over an extensive period of time (*Coles*, *1992*), or had recovered quickly after tropical storm-related losses (*Davies, McKenna & Rasheed*, *2013*). All sites where dugong feeding data were collected had sand with shell sediment and all showed evidence of dugongs feeding (excavated feeding trails) on above and below ground seagrass biomass (Fig. 2). Dugongs have been recorded cropping on seagrass leaves and not leaving any obvious feeding trail (*Preen*, *1992*). However, there was no evidence of cropping at our sites and cropping is not common in soft sediment seagrass meadows (*Johnstone & Hudson*, *1981*; *Preen*, *1992*).

## Data collection

Seagrass meadow characteristics and dugong feeding trails were measured at low tides during the senescent period between May and August 2012. Dugong feeding trail data were collected fortnightly at Double Island and Cockle Bay and monthly at Pallarenda; the Pallarenda site was inaccessible during neap tides. The meadows sampled were similar in species composition and biomass and representative of the region (*Coles*, *1992*; *Davies, McKenna & Rasheed*, *2013*; *McKenzie, Collier & Waycott*, *2012*). All data collection were authorised under 'Marine Parks Permit (G13/36179.1)' and 'General Fisheries Permit (168652).'

To measure the total seagrass removed by foraging dugong, it was necessary to measure the unseen below ground component of the seagrass plant: the roots and rhizomes. This was undertaken by determining whether a linear relationship was present between the above ground (leaves and stems) and the below ground component of the seagrass species. Above and below ground seagrass samples were collected haphazardly across all sites, using a cylindrical pipe corer measuring 150 mm in diameter. Below ground components were sampled to a depth of 100 mm, ensuring the inclusion of all below ground plant biomass likely to be excavated by a dugong (*Anderson & Birtles*, *1978*; *Heinsohn et al.*, *1977*; *Preen*, *1992*). Samples were collected for each seagrass species present at each site, with a minimum of 12 samples collected per site (*Triola & Triola*, *2006*). Samples were washed, epiphytes removed and plants separated into above and below ground components before drying at 40 °C for 48 h or until a stable/consistent dry weight was reached. Dry weights were measured separately for above and below ground components of the plant to generate an equation to estimate below ground weights from above ground weights, when only above ground biomass estimates were available.

Digestible nitrogen was determined for each species that were present at the three foraging sites, and one non-foraging site, to create a predictive hierarchy. Nitrogen samples were collected haphazardly within the four different seagrass meadows. Samples for each species within a site were pooled to obtain the minimum dry weight necessary for nitrogen extraction. Nitrogen weight (milligrams) per seagrass weight (grams dry weight) was determined colorimetrically by the salicylate-hypochlorite method (*Baethgen & Alley*, *1989*). Total nitrogen concentrations were measured as the mean of three absorbance runs, read at 655 nm.

Dugong feeding trail measurements were from 1 m$^2$ quadrats randomly selected within a larger 50 × 50 m plot located where feeding trails occurred; a different plot was sampled each trip. Random selection of 1 m$^2$ quadrats at each site was continued until a minimum of 10 feeding and 10 non-feeding quadrats were obtained for each sampling trip. In each 1 m$^2$ quadrat, species-specific seagrass cover and above ground biomass was estimated using a visual estimation method (*Mellors*, *1991*). To estimate the quantity of seagrass removed by dugongs, we estimated (using the *Mellors* (*1991*) method) the biomass of seagrass within the quadrat immediately adjacent to the trail, equivalent in area to that of the trail. Although dugongs are likely to leave some plant material behind when feeding, in our experiment it was assumed for comparative purposes that whole plants were removed (*Anderson & Birtles*, *1978*; *Preen*, *1992*). The length and width of each feeding trail was recorded to ±0.5 cm. Three measurements of width were taken, one at each end of the trail and one in the middle, to calculate a mean width. Species-specific above ground biomass for each feeding trail was calculated using the feeding trail surface area and the value of the corresponding proportion of above ground biomass for each of the seagrass species.

To avoid damage to the site by destructive sampling, the total biomass available and that removed by dugongs were calculated from above ground biomass estimates (*Mellors*, *1991*) using derived regression equations from the core samples; equations were applied for each species at each site. Data collected on *Cymodocea serrulata* and *C. rotunda* were

combined into *Cymodocea spp.*, as these two species are morphologically similar (*Green & Short*, *2003*; *Waycott et al.*, *2004*).

## Statistical analysis

Data were transformed using log or square root transformations to ensure normality and homogeneity of variances. Analyses of covariance (ANCOVA) were performed to determine whether the relationship between above ground to below ground plant biomass differed across sites for *H. ovalis* and *H. uninervis*; with above ground biomass (g/DW m$^2$) as the independent, below ground plant biomass (g/DW m$^2$) as the dependent, and site as the covariate variable. Where data did not meet the assumptions of a standard regression analysis, a Generalized Least-Squares analysis was used (*Whitlock & Schluter*, *2008*). To determine whether nitrogen differed amongst the dominant seagrass species present (*H. ovalis* and *H. uninervis*), and to test for differences in whole plant nitrogen content within each site, the data was analysed with an ANOVA. A non-parametric test, Wilcoxon rank sum, was used to determine if there were any differences between nitrogen concentrations for *Cymodocea spp.* in the two sites where it was present.

To analyse dugong feeding patterns, a Linear Mixed Effects Model was applied with 'total biomass removed' as the dependent variable. The three intertidal seagrass meadows represented random factors in the analysis, enabling variation between sites to be accounted for during the partitioning of variance. 'Seagrass biomass available' and 'seagrass species present' were entered as fixed factors, and the multiple samples obtained at each site over the four months of data collection were entered as replicates. Standard residual diagnostic tests were performed to determine whether the statistical test was appropriate. To ensure a comparative test for all species present at the feeding sites, an upper limit of 2.80 g/DW m$^2$ was applied to remove outliers from the data.

## RESULTS

### Estimating below ground biomass for seagrass species

There was a significant positive relationship between above and below ground plant biomass for most seagrass species across all sites (*Cymodocea spp., H. ovalis, H. uninervis* and *Syringodium isoetifolium*) (see Supplemental Information 1). Negative y-intercepts for some species suggest those equations are poor predictors of below ground biomass when above ground biomass was low (see Supplemental Information 2). *Thalassia hemprichii* had no significant relationship between above and below ground biomass (see Supplemental Information 1). The below ground component of this plant is deeper than 100 millimetres, (*Green & Short*, *2003*; *Waycott et al.*, *2004*) so all the below ground component would not have been included in our collections. In the analysis of site-specific effects on the two predominant species, *H. ovalis* and *H. uninervis*, the relationship between above and below ground biomass was significantly different among sites (ANCOVA: *H. ovalis*: $F_{4,102} = 21.54$, $p => 0.001$; *H. uninervis*: $F_{5,131} = 14.93$, $p => 0.001$, see Supplemental Information 1).

### Nitrogen concentration of seagrass species

*H. ovalis* and *H. uninervis* had the highest median nitrogen concentrations of the five species at each site. The nitrogen concentration found in *H. ovalis* and *H. uninervis* were not

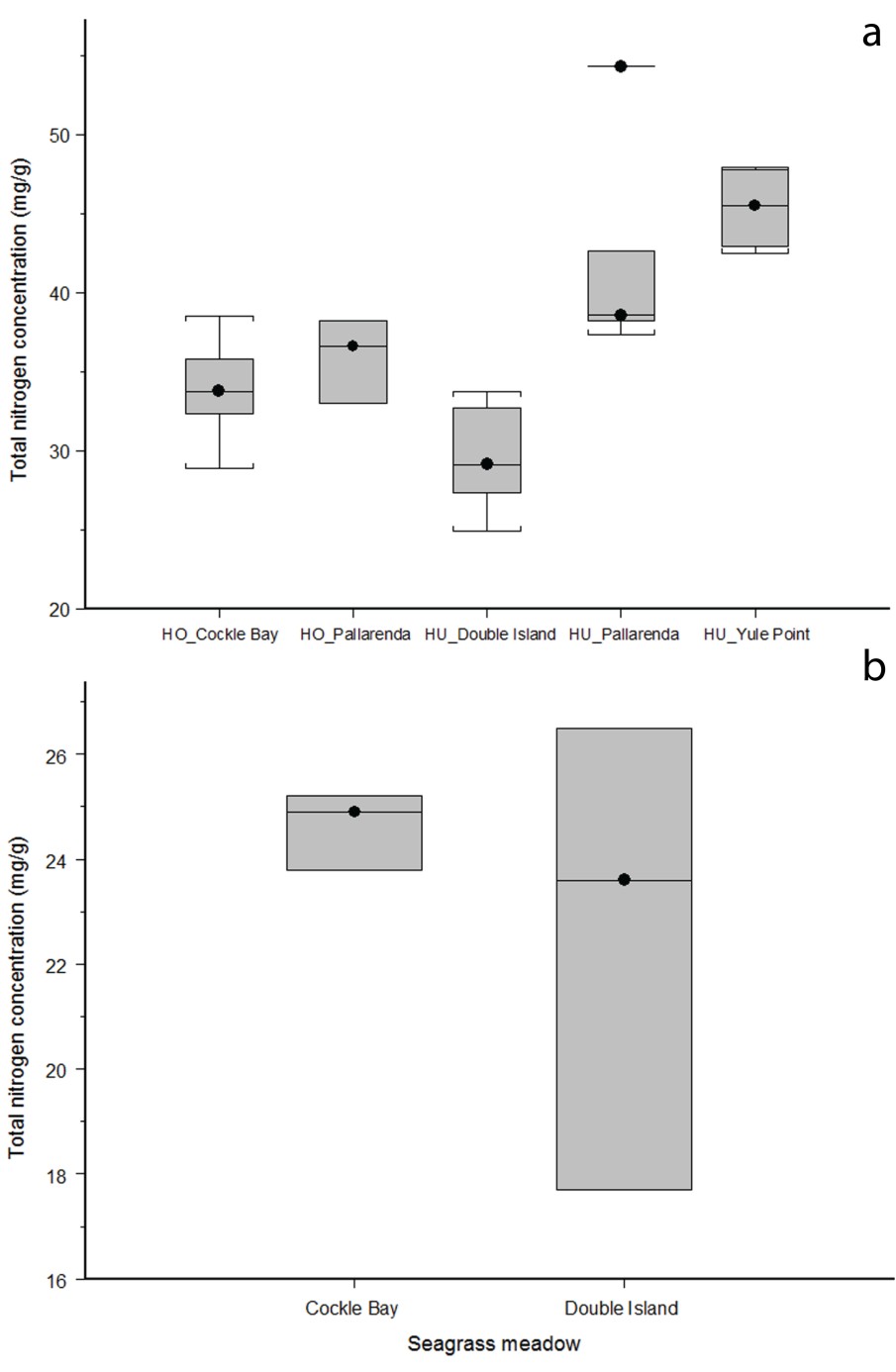

**Figure 3** Whole plant nitrogen weight (mg/g) for seagrass species across different seagrass meadows along the Great Barrier Reef, north-east Queensland Australia; (A) *Halodule uninervis* (HU) and *Halophila ovalis* (HO) are not significantly different between species (ANOVA: $F = 0.29005_{1,3}$, $p = 0.59$), however nitrogen varied significantly across sites (ANOVA: $F = 31.94_{,3}$, $p = \;>0.001$); (B) *Cymodocea spp*. did not differ in nitrogen concentration across the two sites it was present (Wilcoxin rank sum: $Z = 12$, $n = 3$, $p = 0.70$).

**Table 2 Means and standard error of nitrogen weight of seagrass weight (mg/g) for the above ground (leaf and leaf sheaths), below ground (roots and rhizomes) and whole plant of seagrass species within six intertidal seagrass meadows along the Great Barrier Reef, north-east Queensland, Australia.** Samples collected during March to July 2012.

| Site | Above ground | | Below ground | | Total plant | |
|---|---|---|---|---|---|---|
| **Double Is.** | | | | | | |
| *Cymodocea serrulata* | 16.2 | ±2.2 | 3.9 | ±0.5 | 22.6 | ±2.6 |
| *Halophila ovalis* | ID | | ID | | ID | |
| *Halodule uninervis* | 23.0 | ±2.3 | 6.5 | ±0.9 | 29.5 | ±2.8 |
| *Syringodium isoetifolium* | 20.1 | ±0.6 | 9.4 | ±0.3 | 29.5 | ±0.9 |
| *Thalassia hemprichii* | ID | | ID | | ID | |
| **Yule Point beach** | | | | | | |
| *Halophila ovalis* | ID | | ID | | ID | |
| *Halodule uninervis*[a] | 39.8 | ±4.2 | 7.0 | ±1.6 | 46.9 | ±2.2 |
| **Cooya beach** | | | | | | |
| *Enhalus acoroides*[b] | 19.8 | ±2.8 | 11.7 | ±2.0 | 31.7 | ±5.0 |
| *Halophila ovalis* | ID | | ID | | ID | |
| *Halodule uninervis* | ID | | ID | | ID | |
| *Zostera muelleri* | ID | | ID | | ID | |
| **Cockle Bay, Magnetic Is.** | | | | | | |
| *Cymodocea serrulata* | 18.8 | ±0.5 | 5.9 | ±0.4 | 24.6 | ±0.4 |
| *Halophila ovalis* | 27.5 | ±3.0 | 6.4 | ±0.7 | 33.9 | ±3.0 |
| *Halodule uninervis* | ID | | ID | | ID | |
| *Thalassia hemprichii* | 19.9 | ±2.4 | 8.1 | ±0.3 | 27.9 | ±2.8 |
| **Cape Pallarenda** | | | | | | |
| *Halophila ovalis* | 29.1 | ±1.9 | 6.8 | ±0.4 | 35.9 | ±1.5 |
| *Halodule uninervis* | 34.6 | ±7.0 | 7.0 | ±1.6 | 41.6 | ±8.1 |
| **Bowling Green Bay** | | | | | | |
| *Halodule uninervis* | ID | | ID | | ID | |
| *Zostera muelleri* | 25.6 | ±3.1 | 7.0 | ±3.7 | 32.6 | ±5.2 |

**Notes.**
ID, Insufficient data.
[a] Above ground results based on 5 samples; below ground and total results based on 4 samples.
[b] Above ground results based on 11 samples; below ground and total results based on 10 samples.

significantly different (ANOVA: $F = 0.29005_{1,3}$, $p = 0.59$, Fig. 3A), however nitrogen varied significantly across sites (ANOVA: $F = 31.94_{,3}$, $p => 0.001$, Fig. 3A); with *H. uninervis* having higher nitrogen concentrations in Yule Point and Pallarenda. *Cymodocea spp.* did not differ in nitrogen concentration across the two sites where it was present (Wilcoxin rank sum: $Z = 12$, $n = 3$, $p = 0.70$, Fig. 3B). Total nitrogen concentrations varied among species and sites (Table 2). The hierarchal order of whole plant nitrogen concentration (from highest to lowest) for the five seagrass species sampled across all sites was:

*Halophila ovalis = Halodule uninervis > Syringodium isoetifolium > Thalassia hemprichii > Cymodocea spp.*

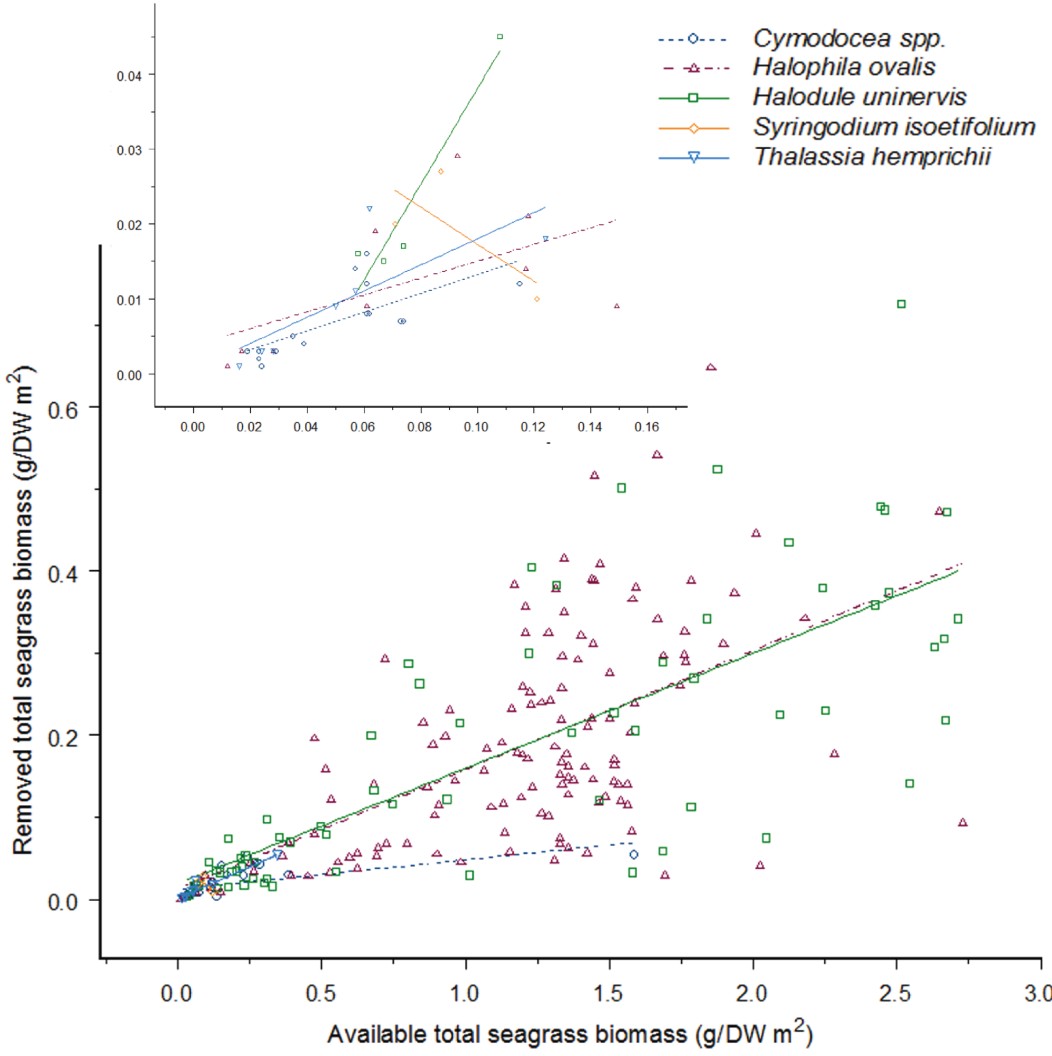

**Figure 4** **As the available seagrass biomass (g/DW m$^2$) increases the amount of seagrass biomass consumed by dugong's increases proportionally for four seagrass species.** There is no difference between feeding on *Cymodocea spp.* to *Halophila ovalis*, *Halodule uninervis* and *S. isoetifolium* (*H. ovalis*: Linear Mixed-Effect model fit by REML: $t = -0.88013$, $p = 0.3797$; *H. uninervis*: Linear Mixed-Effect model fit by REML: $t = -1.01145$, $p = 0.3129$; *S. isoetifolium*, Linear Mixed-Effect model fit by REML: $t = -0.63097$, $p = 0.5287$); however, *Thalassia hemprichii* had greater feeding as biomass increased (Linear Mixed-Effect model fit by REML: $t = 2.06871$, $p = 0.0397$); the $y$-axis has been back-transformed for graphical purposes, and the insert graph is a close up of data points from 0.00 to 0.12 g/DW m$^2$ seagrass biomass available. Data was taken between May to August 2012 from three intertidal seagrass meadows in the Great Barrier Reef, north-east Queensland, Australia; Double Island, Cape Pallarenda and Cockle Bay at Magnetic Island.

## Dugong feeding patterns

With the exception of *S. isoetifolium*, each seagrass species was consumed in increasing amounts as the available biomass of that species increased (Linear Mixed-Effect model fit by REML, Biomass: $F_{1,223} = 373.6722$, $p > 0.001$; Fig. 4). However, significant differences were observed in the rate at which each of the species was consumed at a given biomass (Linear Mixed-Effect model fit by REML, Species: $F_{4,223} = 10.5534$, $p > 0.001$; Fig. 4).
There was no significant interaction effect in this analysis (Linear Mixed-Effect model fit by REML, Biomass*Species: $F_{4,223} = 1.6517$, $p = 0.1623$; Fig. 4), indicating that no seagrass species was consumed at a relatively greater rate when other seagrass species were being consumed. Pair-wise comparisons between species showed that *Cymodocea spp., H. ovalis, H. uninervis*, and *S. isoetifolium* were consumed at equivalent rates per available biomass, while *T. hemprichii* was consumed at greater amounts for a given available biomass (Fig. 4). Sample sizes for both *S. isoetifolium* and *T. hemprichii* were small relative to those for the other species and so analyses and interpretations for these seagrasses species are less reliable. These results did not change for *Cymodocea spp., H. ovalis* and *H. uninervis* when *S. isoetifolium* and *T. hemprichii* were removed from the analysis.

## DISCUSSION

Our study demonstrated that in the Cairns and Townsville regions of tropical north Queensland, Australia, the ingestion of seagrass species by feeding dugongs increased in proportion to their availability for four of the five species, and that these species were consumed at equivalent rates given their available biomass. This indicates that feeding patterns at our sites were influenced most strongly by the available plant biomass and only to a lesser degree by species composition and/or by nitrogen content. Dugongs did not selectively feed on higher nitrogen content species, or in lower biomass areas, as they do elsewhere (*Preen*, *1995*; *Sheppard, Lawler & Marsh*, *2007*). This result was further supported by less feeding trails in other low biomass seagrass meadows close to the study sites (S Tol, pers. obs., 2012). *H. ovalis* and *H. uninervis* were found to be the most common seagrass species available at all sites and were also the species with the highest nitrogen content. This made it difficult to separate feeding targeted at increased biomass from feeding that may have been targeted at high nitrogen content. However, greater feeding on *T. hemprichii*, a species with a lower nitrogen content, further supports our contention that nitrogen plays a lesser role in influencing feeding site use than biomass.

Overall, our results imply that biomass *per se* is the most important factor determining dugong feeding patterns in tropical north Queensland and that feeding at our sites occurs in high biomass locations, independent of species composition or nitrogen content. This finding agrees with dugong mouth and stomach content analyses and feeding trail observations in the Torres Strait and south-east Asian regions of the tropics, where biomass was also considered to be a key factor determining feeding behaviour (*Adulyanukosol & Poovachiranon*, *2006*; *André, Gyuris & Lawler*, *2005*; *Aragones*, *1994*; *Johnstone & Hudson*, *1981*). However, nitrogen concentrations of seagrasses in the Torres Strait were not significantly different among species (*Sheppard et al.*, *2008*), and foraging dugongs would not need to select for higher nitrogen. This effect in the Torres Strait was confounded by differences in digestibility of seagrasses (*Sheppard et al.*, *2008*). Digestibility was not measured in our study, but it is unlikely to be an effect as the most digestible seagrass species (*H. ovalis*) (*Sheppard et al.*, *2008*) was not the most selected species at our study sites.

Intertidal seagrass meadows dominated by *H. ovalis* and/or *H. uninervis* are considered preferential sub-tropical feeding meadows, due to the greater nitrogen content of these

species (*Lanyon*, *1991*; *Mellors, Waycott & Marsh*, *2005*; *Preen*, *1998*; *Sheppard et al.*, *2010*). Our results contradict this theory, suggesting that in eastern Australia dugong feeding behaviour changes with latitude. Regional differences along the Queensland coast in the choice of seagrass species for food and feeding behaviour is likely, as the dugong population is distributed along a wide latitudinal range. Dugongs have shown seasonal changes in their feeding characteristics when living at the edge of their range (*Anderson*, *1994*; *Anderson*, *1998*; *Sheppard et al.*, *2006*), suggesting that feeding behaviour may also have location/site and seasonal influences. The lack of consistent feeding preference associated with seagrass species, biomass, digestibility and/or nutrients in other studies (*André, Gyuris & Lawler*, *2005*; *Johnstone & Hudson*, *1981*; *Preen*, *1995*; *Sheppard et al.*, *2006*), emphasises this potential flexibility and so suggest that our results are relevant only in the tropics and/or at local and regional scales.

Dugong feeding trails are the best non-invasive evidence of feeding, and are common in many tropical intertidal regions, including the GBR and sub-tropical locations (such as Morton Bay and Hervey Bay in south-east Queensland and Shark Bay on the western coast of Australia) (*Aragones*, *1994*; *De Iongh et al.*, *2007*; *Preen*, *1995*). These mostly nearshore sites are also the preferred locations for boating, fishing, hunting and coastal infrastructure development (*Grech, Coles & Marsh*, *2011a*), and dugongs may actively avoid these seagrass meadows for the safety of lower biomass sub-tidal meadows (*Brownell et al.*, *1981*; *Wirsing, Heithaus & Dill*, *2007*). Many seagrass meadows are identified as being at 'high risk' from anthropogenic factors due to increased coastal development (*Grech, Coles & Marsh*, *2011a*; *Orth et al.*, *2006*) and the health and protection of our near-shore coastal seagrass meadows is vital for their survival. The decline in seagrass cover in late 2010 and early 2011 along the tropical east Queensland coast, especially between Cairns and Townsville and their patchy recovery (*McKenna et al.*, *2015*), has emphazised the importance of a better understanding of the characteristics of meadows being grazed by dugong at local and regional scales. Currently in the GBR there are no areas specifically set aside where seagrass meadows are protected for dugong feeding, either from direct impacts or from the influence of adjacent coastal processes. Our research identifies areas in the Cairns to Townsville region where there have been losses in recent times, and where greater levels of protection would be desirable. Advice at appropriate scales to Marine Park and coastal planners that is tailored to regional specific needs, including identifying which areas of seagrass are best able to support dugong populations, is vital for the successful protection of both dugong and their seagrass food source.

In our study sites, high biomass seagrass meadows were the preferred feeding grounds, with seagrass species and nitrogen content important only to a lesser degree. Measuring the chemical composition of seagrass plants is time consuming and expensive. In comparison, rapid seagrass survey techniques are readily available to estimate area, biomass and species composition of meadows (*Short & Coles*, *2001*). The implications of our results, at least for Australian tropical regions, is that these existing rapid seagrass survey techniques can be effectively used to locate and prioritize dugong seagrass feeding grounds for inclusion in management plans. Dugongs are almost obligate feeders on seagrasses and an optimal conservation approach must include strategies to protect both the animals as well as their

food source (*Marsh, O'shea & Reynolds III*, *2011*), and these approaches may have to be regionally specific to be effective.

## CONCLUSIONS

Studying dugong feeding trails are a useful non-invasive tool in understanding feeding behaviour. We found in our tropical sites high biomass seagrass meadows were the preferred feeding meadows, with seagrass species and nitrogen only important to a lesser degree. Management protection and conservation planning for dugongs and dugong seagrass feeding resources needs to be regionally specific for tropical regions. Existing rapid seagrass survey techniques could effectively and economically use seagrass biomass as a surrogate for rapid identification of important dugong feeding meadows for protection in tropical Australia.

## ACKNOWLEDGEMENTS

Advice provided by Helene Marsh, Will Edwards and Paul York at James Cook University, and the reviewers was greatly appreciated and helped to improve the quality of this paper.

### Funding
The authors received no funding for this work.

### Competing Interests
The authors declare there are no competing interests.

### Author Contributions
- Samantha J. Tol conceived and designed the experiments, performed the experiments, analyzed the data, contributed reagents/materials/analysis tools, wrote the paper, prepared figures and/or tables, reviewed drafts of the paper.
- Rob G. Coles and Bradley C. Congdon conceived and designed the experiments, wrote the paper, reviewed drafts of the paper, financial contributions (organisation).

### Field Study Permissions
The following information was supplied relating to field study approvals (i.e., approving body and any reference numbers):
Fisheries Act 1994 - General Fisheries Permit 168652
Marine Parks Permit G13/36179.1.

### Data Availability
The raw data has been supplied as a Supplemental Dataset.
Regression graphs and results used for predicting the below ground biomass from the above ground biomass of seven different seagrass species (Appendix S1 and S2) are

available online. The authors are solely responsible for the content and functionality of these materials. Queries should be directed to the corresponding author.

## Supplemental Information

Supplemental information for this article can be found online at http://dx.doi.org/10.7717/peerj.2194#supplemental-information.

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
