# Peer review of "Dugong dugon feeding in tropical Australian seagrass meadows: implications for conservation planning"

_PeerJ, doi:10.7717/peerj.2194_

## Round 0.1 · original submission · Minor Revisions

· Academic Editor

Minor Revisions

I am sorry for the delay in getting this manuscript back to you, but our last referee was delayed and just got the review back to us now. Overall, all three reviewers are enthusiastic about the value of the work and agree that it should be publishable. The most critical referee feels that the conclusions overstep the strength of the data somewhat because the relative strength of effects of the different variables are tested independently rather than in a single statistical analysis, and the methods are not clear regarding the nitrogen analyses. I can see their point, and feel that it would strengthen the manuscript to address both comments explicitly in the revision. Beyond that criticism, the remainder of the comments are relatively minor corrections or suggestions for clarification that I expect should be simple to address. I am undecided as to whether or not to return it to the final referee for re-evaluation, so I will decide based on the revision and response when I see it whether I feel the need to seek additional input. I look forward to seeing the revised manuscript.

Reviewer 1 ·

Basic reporting

No comments

Experimental design

No comments

Validity of the findings

Sentences in lines 205-6 and 209-212 are a bit confusing. Is there a difference between the two? If yes, then can be be explained with a bit more clarity?

Was dugong herbivory determined as a rate of removal? If not, then consider changing sentences accordingly using either quantity or amount of removal or another appropriate word.

Additional comments

This is a nice piece of work dealing with some aspects of dugong food preferences that are quite difficult to understand in-situ. While I do feel that it would have been good to have more sampling locations, I understand it is not always practical. Nevertheless, the authors presented the work well.

Reviewer 2 ·

Basic reporting

The manuscript fulfilled the journal's standards and criteria for basic reporting.

Experimental design

The manuscript fulfilled the journal's standards and criteria for experimental design.

Validity of the findings

The manuscript fulfilled the journal's standards and criteria for validity of findings.

Additional comments

Please check through the manuscript for consistency of reporting SI unit formats and also scientific names.

The science was sound and rigorous, producing good scientific results. However, the Discussion section needs some rearranging for a better flow - please see the detailed comments made in the manuscript PDF (mouse over the "Discussion" heading to see the comments).

Please also review your manuscript and ensure that you do not introduce phrases or concepts in your Discussion that was not previously mentioned earlier in the manuscript OR ensure that these are mentioned appropriately earlier on in the manuscript if you are to bring them up again in the Discussion (e.g. rapid seagrass survey techniques). Please refer to manuscript PDF on this matter.

As the manuscript title ends with "implications for conservation planning", I would have liked to see the manuscript wrap up with more details pertaining to conservation planning for the study site rather than general and "overview" style of recommendations for conservation of seagrass.

Annotated reviews are not available for download in order to protect the identity of reviewers who chose to remain anonymous.

Reviewer 3 ·

Basic reporting

The English is clear enough and has sufficient introduction and background. However, I had difficulties in understanding the regression lines in Figures 3 and 6 because of unclear symbology (indistinct symbols and lines when printed). It was especially difficult to spot the symbols for Syringodium isoetifolium and Cymodocea spp. in Figure 6. Possibly, some of the symbols are overlapping and I would suggest adding some jitter to the data points to avoid this problem. I would suggest that the authors use a graphics software for all their figures.

Figure 3 does not serve a direct purpose in answering the research question about what drives dugong feeding rates on seagrass. The figure merely addresses a methodological issue, i.e. how to estimate belowground biomass from aboveground biomass of seagrass. Thus, Figure 3 would be better placed in the appendix or supplementary materials. On that note, the authors pay a large amount of attention to the relationship between aboveground and belowground seagrass biomass, by having Figure 3, a whole paragraph in the results section (lines 176-188), and Appendix 1 dedicated to it. The authors need to consider whether this is indeed the main point in their study. If it is, then they should have also addressed it in the introduction.

The caption for Figure 4 did not reflect the figure properly. It merely compares the median nitrogen weights for 2 species and contrary to the caption, does not make comparisons between seagrass meadows.

Experimental design

I have concerns about the way the authors have designed their sampling plan for the nitrogen analysis. They state that they collected seagrass species for nitrogen analysis at foraging sites (lines 125-126) to determine whether nitrogen differed amongst the dominant seagrass species (lines 160-161). However, if their overall objective is to see if nitrogen determined feeding selectivity, then they should have sampled outside foraging sites as well, and looked for differences in nitrogen between each species within and outside a foraging site. The authors can either (a) rewrite the methods section if they did indeed sample within and outside foraging sites, or (b) explain why they have opted to compare nitrogen between species that have already been 'selected' by dugongs.

Validity of the findings

To me, the authors have overreached their data. They concluded about how feeding patterns were influenced “most strongly by the available plant biomass and only to a lesser degree by species composition and/or by nitrogen content”. However, plant biomass, species composition and nitrogen content were not tested in a single statistical model with variance contributions for each variable that would have allowed them to conclude about the relative effects of the different variables. Instead, the variables were tested separately: biomass and species composition in a Linear Mixed Effects Model and nitrogen content in a non-parametric regression model. In the Linear Mixed Effects Model, I cannot tell if they sampled across a range of very low to very high biomass in Figure 6 because the log y-axis variable was not backtransformed. As for the non-parametric regression model for nitrogen content, I have already stated my concerns above. I suggest the authors soften their conclusions about the relative effects of the variables on dugong feeding.

Additional comments

Line 31: Change from ‘predominately’ to ‘predominantly’.
Line 50-52: The authors state that “Research comparing nitrogen content found that intertidal seagrass plants contain significantly more nitrogen than sub-tidal plants (Sheppard et al 2008)…”. I’m not entirely sure that this finding about nitrogen can be attributed to Sheppard et al 2008. The study of Sheppard was located in subtidal meadows (not comparatively across intertidal and subtidal meadows) and although they found that seagrass lignin decreased with depth, they did not indicate the same for nitrogen in that paper.
Line 54-56: To better highlight the main point in this sentence, I would suggest rewording to “In contrast, studies of dugong feeding in other tropical regions regularly observe feeding across all species present, with the exception of Enhalus acoroides”.
Line 56-59: In their statement that “This difference in feeding behavior may be due to seagrass biodiversity being lower in south-east Queensland than in those in the tropics”, the authors suggest that Queensland dugongs seem to be more selective in feeding than dugongs elsewhere because of lower seagrass biodiversity in south-east Queensland. I would have thought that the reverse would be true, that selectivity in feeding is more likely to occur in high biodiversity seagrass meadows because the animals can afford to be selective.
Line 125-126: If samples for nitrogen analyses were collected “at the foraging sites”, does this mean you analysed only seagrass species that were ‘selected’ by dugongs? Sampling within and outside foraging sites and looking for nitrogen differences between individual species within and outside foraging sites, could have better answered your question.
Line 128: Replace ‘colorimetically’ with ‘colorimetrically’
Line 156-158: The explanation for ANCOVA may be improved by specifying the independent, dependent and control variable (covariate) at the outset.
Line 160-163: I suggest rewriting the sentence to read as “To determine whether nitrogen differed amongst the dominant seagrass species present, and to test for differences in whole plant nitrogen content within each site for Halophila ovalis and Halodule uninervis, the Wilcox rank sum and Kruskal-Wallace non-parametric tests were used”.
Line 165: Revise from “Mixed Linear Effects Model to “Linear Mixed Effects Model”.
Line 171-172: It is unclear what the following sentence means: “Available biomass for each seagrass species was analysed across a comparative range of values with an upper limit of 2.8 g/DW m2”.
Line 177: When the authors state “there was a significant positive relationship between above and belowground plant biomass…”, it would be helpful to point the reader to a figure or table that reflects this.
Line 187-188: Do you mean p<0.001 instead of p>0.001? Check that this is also correct for all the significance values in figures, tables and appendices.
Line 192-193: It is unclear what you mean by “The nitrogen concentration found in Halophila ovalis and Halodule uninervis were not significantly different between species among sites”. I suggest removing “between species” since the species names have already been referenced.
Line 215-219: You can do away with the t and p values because it distracts from your main point. It is enough to point the reader to Figure 6 that already contains all these statistical values.
Line 259-261: The authors state “This indicates that feeding patterns at our sites were influenced most strongly by the available plant biomass and only to a lesser degree by species composition and/or by nitrogen content”. I would suggest softening this sentence because nitrogen content, plant biomass and species composition were not tested in a single statistical model. Nitrogen content was tested for differences between seagrass species in a non-parametric model, while biomass and seagrass species were tested in a Linear Mixed Effects Model. Moreover, the feeding rate on seagrass was not regressed on seagrass nitrogen content.
With the conclusion the authors draw about the relative effects of nitrogen, seagrass species and biomass on feeding rates, I would have expected to see a stepwise regression analysis and the amount of variance explained by each variable on feeding rates.

Table 1: The authors should address the following:
1. Explain the NA values
2. Explain the missing values for Zostera muelleri and Thalassia hemprichii.
3. Insert standard deviations/errors alongside the average biomass values.
4. What does ‘Combined above’ mean?
5. Be consistent in the number of decimal places used.

Figure 3: The authors should address the following:
1. Label the figures (a) and (b)
2. The symbols and regression lines were very indistinct when printed out. I’d suggest you use a graphics software.
3. This figure shows belowground biomass regressed on aboveground biomass for the purpose of estimating aboveground biomass, which is a methodological issue. It doesn’t provide much of a purpose in supporting your main question about feeding rates, and may be better placed in the appendix or supplementary materials.
Figure 4: This figure shows the median nitrogen weight per seagrass dry weight, which overlaps with Table 2. However, if you still want to use this figure, I’d suggest rewording the caption to better reflect the figure. In its current form, Figure 4 merely compares the median nitrogen weights for 2 species and contrary to your caption, does not make comparisons between seagrass meadows.

Table 2: Please address the following:
1. Explain the missing standard errors. I assume it is because the values were obtained from the regression equations? If so, this needs to be explained in the table.
2. In the footnote of the table, change ‘above and total results based on x samples’ to ‘aboveground and total plant results based on x samples’.
3. If there were only 2 samples for Thalassia hemprichii, I’d suggest removing this species from the analysis.

Figure 6: Symbols and lines were very indistinct. Considering that this figure is integral to associating seagrass biomass with dugong feeding rates, it would be helpful if you backtransformed the total seagrass biomass removed.

---

## Round 0.2 · accepted · Accept

· Academic Editor

Accept

Having read through your manuscript I am satisfied with the revisions and how you have addressed the referee comments. I am happy to accept your manuscript and move it forward into production.